# Increasing Hypopnea in Sleep Breathing Disturbance Improves Postoperative Oxygen Saturation in Patients with Very Severe Obstructive Sleep Apnea

**Ethan I. Huang [1,2,3,]\***, **Shu-Yi Huang [2,4,5]**, **Yu-Ching Lin [2,3,4,6]**, **Chieh-Mo Lin [4,7]**,
**Chin-Kuo Lin [4,7]**, **Ying-Chih Huang [8]** and **Jian-An Su [9]**

1   Department of Otolaryngology, Chang Gung Memorial Hospital, Chiayi 61363, Taiwan
2   Sleep Center of Chang Gung Memorial Hospital, Chiayi 61363, Taiwan; 8802022@cgmh.org.tw (S.-Y.H.);
    lin0927@cgmh.org.tw (Y.-C.L.)
3   School of Medicine, Chang Gung University, Taoyuan 33302, Taiwan
4   Division of Pulmonary and Critical Care Medicine, Chang Gung Memorial Hospital, Chiayi 61363, Taiwan;
    jamesnomo@gmail.com (C.-M.L.); lingh@cgmh.org.tw (C.-K.L.)
5   Department of Nursing, Chang Gung University of Science and Technology, Chiayi 61363, Taiwan
6   Department of Respiratory Care, Chang Gung University of Science and Technology, Chiayi 61363, Taiwan
7   Graduate Institute of Clinical Medical Sciences, College of Medicine, Chang Gung University,
    Taoyuan 33302, Taiwan
8   Department of Neurology, Chang Gung Memorial Hospital, Chiayi 61363, Taiwan; ngingchi@cgmh.org.tw
9   Department of Psychiatry, Chang Gung Memorial Hospital, Chiayi 61363, Taiwan; sujian@cgmh.org.tw
*   Correspondence: ehuang@alumni.pitt.edu

**Abstract:** In patients of very severe obstructive sleep apnea (OSA) with confined framework, reducing volume is difficult to achieve a postoperative apnea-hypopnea index (AHI) qualifying the classical surgical success. However, a higher AHI with a larger part of hypopneas may have similar or even less severity of oxygen ($O_2$) desaturation, compared to a lower index mostly made of apneas. Here, in 27 consecutive enrolled patients, we show that besides the improvement of mean AHI, the multilevel surgery increased hypopnea in AHI from 29.1% to 77.3%, and improves postoperative $O_2$ saturation by reducing desaturation frequency (mean desaturation index decreased from 62.5 to 24.4 events/h) and level (mean oxyhemoglobin saturation of pulse oximetry ($SpO_2$) desaturation cut down from 10.0 to 5.8%). The mean $SpO_2$ improved from 92.3% to 94.7%, and the improvement was positively related to the proportion increase of hypopnea/AHI. The results suggest that the non-framework surgery could help patients with very severe OSA whose AHIs are ≥60 events/h in terms of improving postoperative $O_2$ saturation. Due to the improvement also presented in those not qualified as classical surgical success, further studies are needed to clarify the connection between $O_2$ desaturation and various consequences to reconsider defining a surgical success.

**Keywords:** palatoplasty; one-stage; comorbidity; retropharynx; maxillomandibular advancement; Continuous Positive Airway Pressure (CPAP)

---

## 1. Introduction

Intermittent $O_2$ desaturation in adult obstructive sleep apnea (OSA) patients has been proven to correlate with possible consequences such as hypertension [1–4], cognitive deficits [5,6], and cardiovascular disorders [7–9] (e.g., pulmonary hypertension [8], coronary artery disease [8,9], cerebrovascular disease [8,9], idiopathic cardiomyopathy [8], and congestive heart failure [8,9]). However, $O_2$ desaturation with different severities may result from similar apnea-hypopnea indexes

(AHIs) and vice versa [10,11]. Studies have shown that deeper and longer $O_2$ desaturation events may contribute more to the daytime sleepiness [12,13] and harmful health consequences than the shallower and shorter events [10]. More severe $O_2$ desaturations come from apneas, compared to hypopneas (e.g., see [11]). Intermittent $O_2$ desaturation results from sleep apneas and hypopneas, which constitute a variety of components in most of the AHI.

A common ground of treating OSA patients could be improving $O_2$ saturation to reduce consequences. However, surgically treating very severe OSA patients (defined to AHI $\geq$ 60 events/h here) can be difficult due to their narrow retroglossal space and confined framework, which is hard to enlarge via conventional uvulopalatopharyngoplasty (UPPP) surgery [14]. In the literature, very severe OSA refers to having a high AHI or respiratory disturbance index (RDI) of 40 [15,16], 50 [17–21], 60 [22–26], 70 [27], or 100 events per hour [28]. It is a distinct subgroup and differs from OSA with other severities. For example, patients with very severe OSA may have a low level (as 77 mmHg) of diurnal partial pressure of oxygen while they are not in sleep [29]. There may be residual on-Continuous-Positive-Airway-Pressure (CPAP) AHIs > 20 events per hour [30]. They have less positional OSA [31], a high prevalence of hypertension [29], higher insulin resistance [32], and about three times the chances of heart block compared to an unselected group of OSA patients [33]. The unfavorable anatomy is not necessarily related to obese or high body mass index (BMI) (e.g., see [23,25]).

The anatomical limitations and the characteristics of this group of very severe OSA make it hard (e.g., Walker's 17% (1/6) [34], Vilaseca's 9% (1/11) [23], and Mickelson's 30% (3/10) [35]) to achieve the commonly used "surgical success" in sleep surgeries, i.e., AHI < 20/hour and AHI reduction > 50%, first proposed by Sher et al. [36]. It is unclear whether increasing hypopnea in AHI, regardless of the qualification of surgical success, improves postoperative $O_2$ saturation and reduces the consequences. It is also unclear whether the one-stage multilevel surgery we proposed for severe OSA without operating on the bone [37] successfully improves the postoperative $O_2$ saturation with an unfavorable anatomical stage.

If more severe $O_2$ desaturations come from apneas than hypopneas [11] and the surgery may shift apneas to hypopneas, it is reasonable to hypothesize that increasing hypopnea in AHI improves postoperative $O_2$ saturation in refractory very severe OSA patients. Here, we compared the pre- and post-operative postoperative polysomnographies (PSGs) and showed how increasing hypopnea in AHI reduces frequency of desaturation (by desaturation index (events/h)), reduces severity of desaturation (by mean $SpO_2$ desaturation (%)), and improves mean $SpO_2$ after the one-stage multilevel surgery in very severe OSA patients. We also showed how the improvement can be seen in classical non-surgical success patients.

## 2. Materials and Methods

Between March 2015 and May 2020, we enrolled consecutive patients with very severe OSA who met these criteria to the study:

- Age $\geq$ 20 years
- Unsuccessful or refusal of CPAP
- AHI $\geq$ 60 events/h
- Received a one-stage multi-level sleep surgery with the modified ZPP performed with one-layer closure and open partial tongue-base glossectomy
- Available preoperative and postoperative PSGs for required recordings, including desaturation index and $SpO_2$.

Patients were not specially selected for surgery by any other criteria unlisted above. There were no specific exclusion criteria. Every PSG was conducted overnight in the level 1 sleep laboratory of the authors' tertiary referral hospital. In a PSG, an apnea is defined as the complete cessation of airflow for at least 10 s. A hypopnea is defined as a decrease in airflow $\geq$ 30% for at least 10 s that is accompanied by either EEG signs of arousal or by a 4% or greater decrease in oxygen saturation. We performed pre-

and post-operative endoscopies to evaluate the anatomy. Patients received general anesthesia by oral intubation to allow routine septomeatoplasty, and breathed through the mouth for 2 to 3 days before we removed the nasal packing. Modified Z-palatoplasty (ZPP) with one-layer closure was performed as illustrated in our previous report [37]. Partial open tongue-base resection was completed with transoral ($CO_2$) laser microsurgery or transoral robotic surgery (TORS) for hypopharyngeal obstruction according to the preoperative endoscopic assessment. After the surgery, we cared for all patients in general ward areas with oximeter monitor. Intravenous Dynastat was prescribed twice daily on patients' request. No intravenous or oral narcotics were given to prevent respiratory depression.

A PSG followed about 6 months (193 ± 67 days) after the surgery, and sleep parameters were considered stable. We measured the mean change in AHI compared to the mean AHI before surgery, AHI reduction, as suggested by Caples, S. M. et al. [38]. In the components of AHI, we then calculated the percentage of hypopnea in AHI and compared the desaturation index, the mean $SpO_2$ desaturation, and the mean $SpO_2$ before and after the surgery. We performed a paired *t*-test to examine the changes in AHI reduction, percentage of hypopnea in AHI, desaturation index, mean $SpO_2$ desaturation, and mean $SpO_2$, against no change after the surgery. We plotted scatter graph to exclude outliers and justify the application of correlation coefficient. We calculated the correlation coefficient to show the relationship between proportion change of hypopnea/AHI (defined here as (postoperative-preoperative)/preoperative fraction of hypopnea in AHI) and change of mean $SpO_2$ (defined here as (postoperative-preoperative)/preoperative mean $SpO_2$). We then tested the change of mean $SpO_2$ in "non-surgical success" patients to see if clinical and statistical improvements can be seen if a classical surgical success could not be qualified. The statistical significance was tested as $\alpha = 0.05$.

All statistical examinations were performed in MATLAB 9.4.0.813654 (MathWorks, Natick, MA, USA).

## 3. Ethical Statements

The Institutional Review Board of Chang Gung Medical Foundation, Taiwan, approved the study methods, protocols, and informed consent of subjects, with relevant guidelines and regulations.

## 4. Results

This study enrolled 27 very severe OSA patients with 24 men and 3 women between 29 and 63 years of age (mean = 47.4; standard deviation (SD) = 10.6). The mean BMI was 28.5 with a standard deviation of 3.5 kg/m$^2$. All patients received ZPP with one-layer closure and partial open tongue-base resection. Twenty-five received routine septomeatoplasty. One patient underwent UPPP at another hospital before visiting our clinic. Three and one patients underwent routine endoscopic sinosurgery and adenoidectomy, respectively.

After the surgery, the mean AHI (with 1 SD; the same for the follows) was reduced from 73.8 ± 10.3 to 33.2 ± 20.4 events/h ($p < 0.001$). The mean AHI reduction was 40.5, with a 95% confidence interval of 31.9 to 49.1 events/h. Figure 1 illustrated the individual AHI reductions. In these AHIs (including central and mixed indexes), the mean obstructive apnea index reduced 35.7 events/h, with a 95% confidence interval of 28.1 to 43.3 events/h ($p < 0.001$) (Figure 2). The hypopnea index almost remained the same, with a small change of 2.2 events/h ($p = 0.6562$) (Figure 3). The mean portion of hypopnea in AHI increased from 29.1 ± 23.9 to 77.3 ± 25.1% ($p < 0.001$) (Figure 4). The mean increase was 48.2%, with a 95% confidence interval of 37.7 to 58.8%. The mean desaturation index decreased from 62.5 ± 12.5 to 24.4 ± 16.3 events/h (mean reduction was 38.1, with a 95% confidence interval of 30.6 to 45.6 events/h, $p < 0.001$) (Figure 5), and the mean $SpO_2$ desaturation was cut down from 10.0 ± 5.0 to 5.8 ± 1.9% (mean reduction was 4.2%, with a 95% confidence interval of 2.3 to 6.1%, $p < 0.001$) (Figure 6). The mean $SpO_2$ improved from 92.3 ± 3.4 to 94.7 ± 1.5% (mean improvement was 2.4%, with a 95% confidence interval of 1.0 to 3.8%, $p = 0.0014$) (Figure 7). A correlation of the data revealed that the improvement of mean $SpO_2$ was positively related to the proportional increase of hypopnea/AHI,

r = 0.483, *p* = 0.011, two-tailed test (Figure 8). However, the Spearman correlation was 0.374, *p* = 0.0557, showing the probable limitations of small case number or existence of outliers.

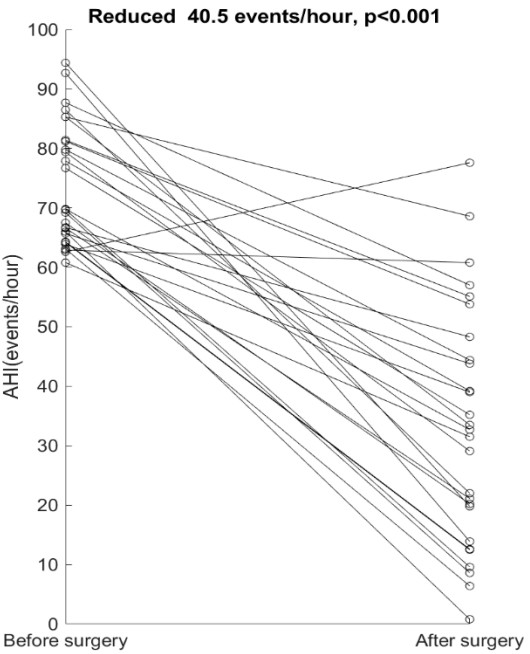

**Figure 1.** Individual apnea-hypopnea-index (AHI) changes revealed a significant reduction after the surgery.

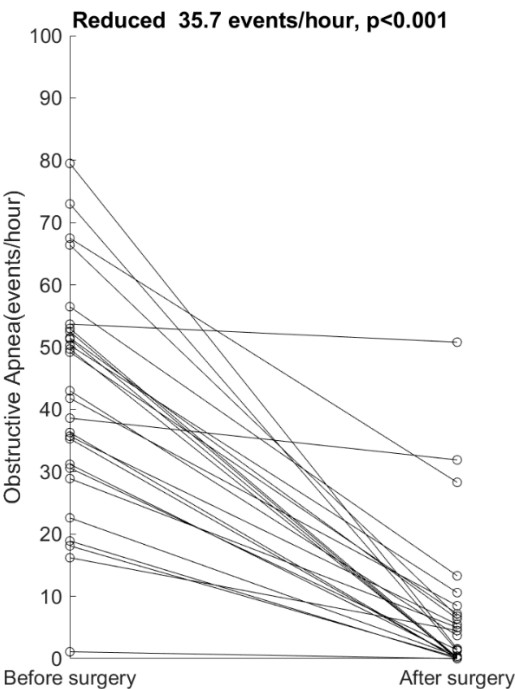

**Figure 2.** Individual obstructive apnea index changes revealed a significant reduction after the surgery.

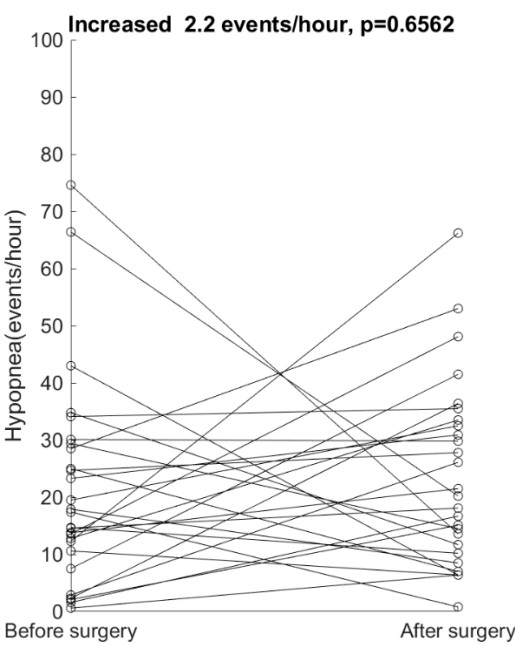

**Figure 3.** Individual hypopnea index did not reveal a significant change after the surgery.

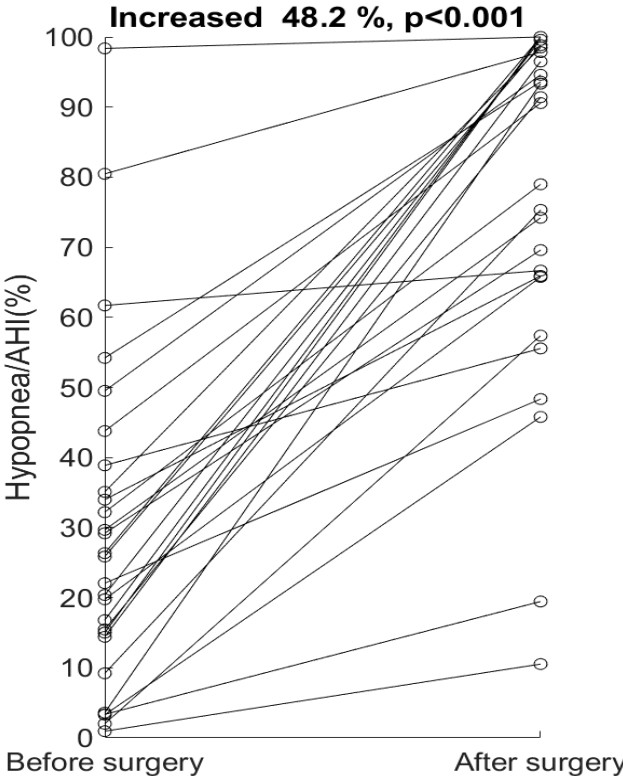

**Figure 4.** Individual portion change of hypopnea in AHI. After the surgery, the mean hypopnea portion increased 48.2%, $p < 0.001$.

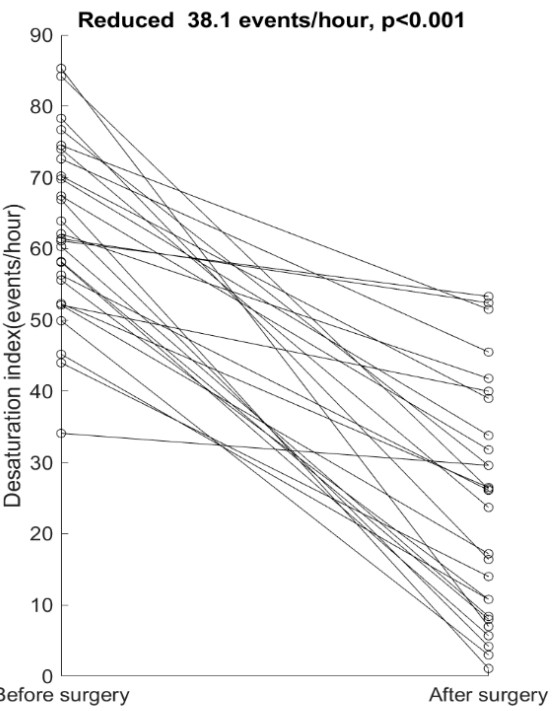

**Figure 5.** Individual change of desaturation index. After the surgery, the mean desaturation frequency cut 38.1 events/h, $p < 0.001$.

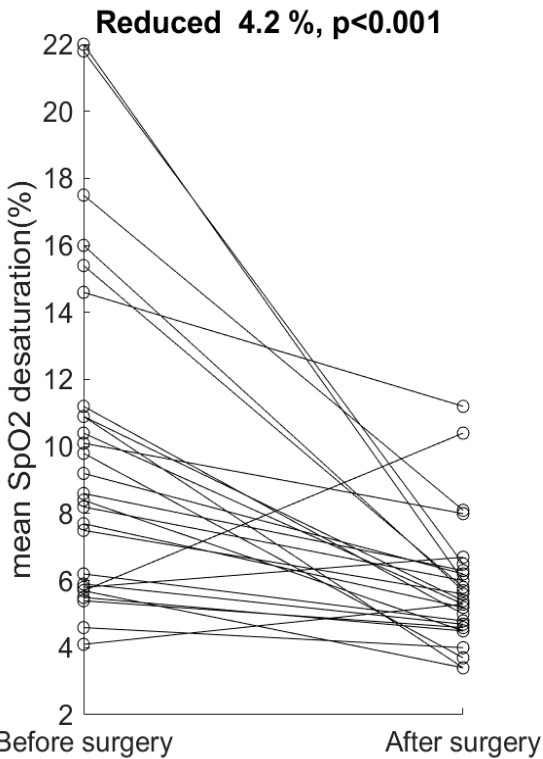

**Figure 6.** Individual change of mean $SpO_2$ desaturation. After the surgery, the mean desaturation severity reduced 4.2%, $p < 0.001$.

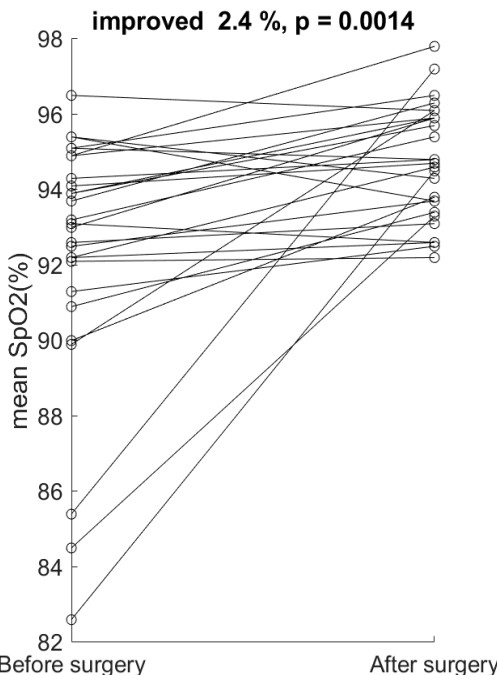

**Figure 7.** Individual change of mean $SpO_2$. After the surgery, the mean $O_2$ saturation improved 2.4%, $p = 0.0014$.

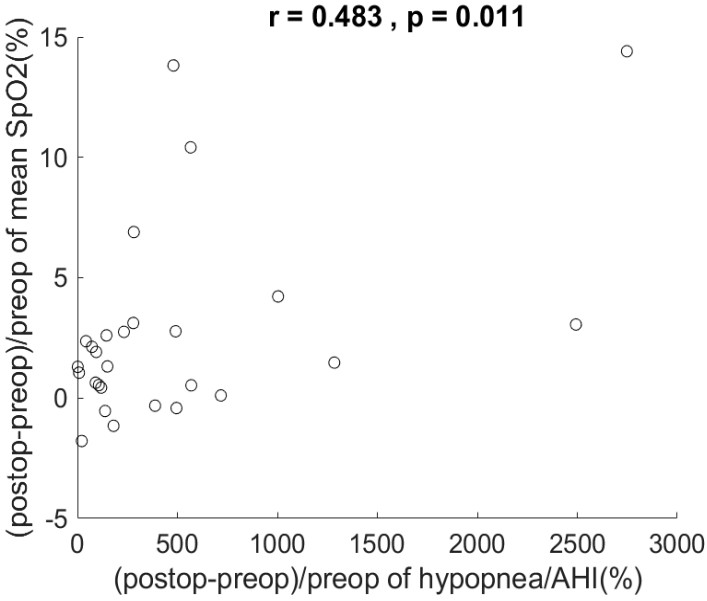

**Figure 8.** Scatter plot and correlation of proportion change of hypopnea/AHI (defined here as (postoperative-preoperative)/preoperative fraction of hypopnea in AHI) vs. change of mean $SpO_2$ (defined here as (postoperative-preoperative)/preoperative mean $SpO_2$). It shows that the improvement of mean $SpO_2$ was positively related to the proportional increase of hypopnea/AHI.

Nineteen out of 27 patients did not qualify the classical surgical success because each had a postoperative AHI > 20 events/h, although all had AHI reductions > 50%. In these 19 "non-surgical success" patients, the mean $SpO_2$ improved from 91.7 ± 3.7 to 94.3 ± 1.6% ($p = 0.0087$) (Figure 9). Among the remaining eight patients, the AHI reduced from 73.0 ± 13.0 to 10.5 ± 5.6 events/h and qualified as "surgical success". However, the improvement of mean $SpO_2$ from 93.8 ± 2.0 to 95.7 ± 0.8% did not show statistical significance ($p = 0.0625$).

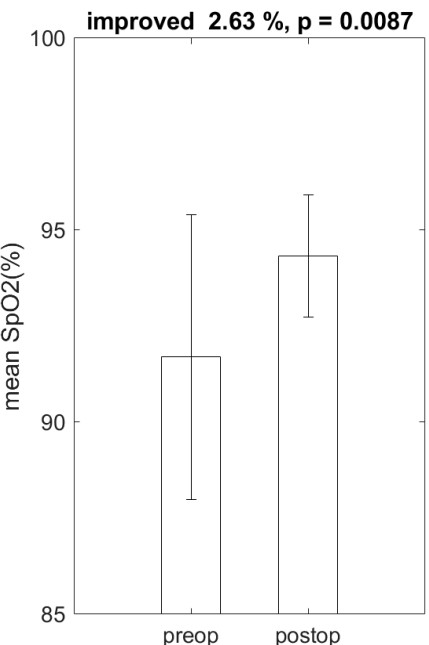

**Figure 9.** Mean SpO$_2$ improved 2.63% in "non-surgical success" patients (19/27), *p* = 0.0087.

## 5. Discussion

The results show that the multilevel surgery shifted (Figure 4) apneas that cause more severe O$_2$ desaturations (Figure 2) to hypopneas that allow some passing airflow (Figures 3 and 4). The surgery increased the portion of hypopnea in AHI. Although we believe apnea reduction is the key to improve O$_2$ saturation, apnea index after the surgery easily reaches the floor (floor effect, i.e., close to 0, see Figure 2) and distorts the pre- and post-operative comparison. That is why we selected hypopnea/AHI ratio as the measurement. The improvement of postoperative O$_2$ saturation in very severe OSA patients was seen in several ways: 1. Reduced desaturation frequency by desaturation index; 2. Reduced desaturation severity by mean SpO$_2$ desaturation; and 3. Improved mean SpO$_2$.

The surgery improved mean SpO$_2$ in the 19 "non-surgical success" patients, regardless of the qualification of surgical success. Although the improvement of mean SpO$_2$ in the eight "surgical success" cases did not reveal statistical significance (*p* = 0.0625), it needs to be aware that this could be the limitations of a small number or the ceiling effect—these 8 patients had higher preoperative mean SpO$_2$ of 93.8%, and it is harder to get better than 95% after the surgery.

Classical surgical success—AHI < 20 events/h and AHI reduction > 50%, first proposed by Sher et al. [36]—was frequently reported in associated studies with less severity and might serve as an outcome measure for comparison. The surgical success of the present study was 29.6% (8/27), which was worse than Mickelson's 30% (3/10) [35] but better than Walker's 17% (1/6) [34] and Vilaseca's 9% (1/11) [23] (calculated from detailed subject information provided in available literature for non-framework sleep surgeries in very severe OSA patients). The mean AHI reduction in the present study was 40.5 events/h, which was worse than Mickelson's 42.2 events/h [35] but better than Walker's 38.2 events/h [34] and Vilaseca's 14.4 events/h [23].

If improving O$_2$ saturation to reduce consequences is the goal, we may need to redefine "success of surgery", at least for very severe OSAs. The results suggest that factors evaluating surgical success could include a combination of AHI reduction and hypopnea ratio, desaturation index, mean SpO$_2$ desaturation, and mean SpO$_2$. Future studies are needed to clarify the connection between the O$_2$ desaturation, including level and frequency, and various consequences. These results may intrigue future works to investigate a better classification of surgical success, taking into account initial, but not only, AHI.

## 6. Conclusions

Besides AHI reduction, the multilevel surgery in this study increased hypopnea in AHI from 29.1% to 77.3%, and improves postoperative $O_2$ saturation by reducing desaturation frequency and level in very severe OSA patients. In other words, the multilevel surgery not only reduced the total number of sleep breathing disturbance per hour, but also shifted apneas that cause more severe $O_2$ desaturations to hypopneas that allow some passing airflow. The mean desaturation index decreased from 62.5 to 24.4 events/h, and the mean $SpO_2$ desaturation cut down from 10.0% to 5.8%. If improving $O_2$ saturation to reduce consequences is the goal and the key for surgical success, the results suggest that the surgery could benefit patients with very severe OSA whose AHIs are ≥60 events/h in terms of improving postoperative $O_2$ saturation. Since the improvement of $O_2$ saturation also presented in those cases not qualified classical surgical success, we may need to reconsider the definition of surgical success, at least in very severe OSA patients.

**Author Contributions:** E.I.H. designed the study, collected data, read the PSGs, conducted statistical analysis, and wrote the first draft of this manuscript. S.-Y.H. and Y.-C.L. enrolled the patients, helped to interpret and present the data, and made critical comments on this manuscript. C.-K.L., C.-M.L., Y.-C.H., and J.-A.S. enrolled the patients, analyzed the data, and revised the manuscript. All authors have read and agreed to the published version of the manuscript.

**Funding:** This research received no external funding.

**Conflicts of Interest:** The authors declare no conflicts of interest.

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
