# Peer review of "Increasing Hypopnea in Sleep Breathing Disturbance Improves Postoperative Oxygen Saturation in Patients with Very Severe Obstructive Sleep Apnea"

_applsci, doi:10.3390/app10186539_

Round 1
Reviewer 1 Report
- This study shows the effect of multilevel surgery on severe OSA, but it is a single arm retrospective study, so the position of this treatment is not clear. The position of multilevel surgery will become clearer when other treatments and their effects on the target patients and data on the natural history of severe OSA are added to the study. More reviews of previous studies should be added.
-
There are some outliers in Figure6. Please check if the Spearman correlation coefficient shows the same trend.
- In the Results section,“Error! Reference source not found." can be found everywhere. Please refer to it properly.
-
In addition to the point estimate and p-value, the 95% confidence interval should also be shown when expressing the treatment effect.
Reviewer 2 Report
I reviewed the manuscript entitled “Increasing hypopnea in restricted breathing disturbance improves postoperative oxygen saturation in patients with very severe obstructive sleep apnea”.The work is well written and clarifies the results clearly.
However, I have found major limitations.
Title
Authors have used the term”very severe obstructive sleep apnea”. The AASM divides OSA into mild, moderate and severe OSA. There is no „very severe OSA” grade according to AASM.
Moreover, autors used term” restricted breathing disturbance” . The „sleep breathing disorder” would be more appropriate, or just „obstructive sleep apnea” could be used. Term “restriction” in pulmonology means low forced vital capacity (FVC) with a normal FEV1/FVC ratio. Confusing.
Methods
The methods section is very poor.
The group is not numerous and heterogeneous (non-surgical success” patients (n=19), surgical success (n=8) . The control group is missing.
Lack of parity (24 men and 3 women) restricts study interpretation.
The enrolled patients have undergone various surgical procedures (UPP, septomeatoplasty, sinosurgery, adenoidectomy) that may affect the results of the study.
No exclusion criteria
The polysomnographic data are poor.
Where was PSG conducted? home or in sleep lab? single-night? Criteria of PSG assesment are missing.
The basic polysomnographic parmeters are missing: sleep latency, efficiency, arousals, sleep architecture, heart rate, AI, CAI, OAI, MAI, HI, Cheyne-Stokes breathing,
The information that PSG followed 6 month after surgery should be in “methods”, not in “results”.
It seems that probably authors used respiratory polygraphy, not polisomnography.
Results
SpO2 Duration < 90% and minimal saturation are missing.
The procentage of hypopneas is insufficient parameter. The authors assess hypopneas but there is no information about hypopnea index (HI) !, which is the most important polysomnographic parameter regarding hypopneas. There is inforamation about central apneas or central hypopneas.
The apnea aindex (AI) should be also presented and compared berfore and after surgery.
The studied group is very unusual . The high severity of OSA (AHI-73) was accompanied by relatively low BMI (28.5) . It should be disscussed.
The correlation between HI reduction and respiratory indices reductions should be presented.
The correlation between AI reduction and respiratory indices improvment should be also presented. The respiratry indices improvement may be also related to apneas index reduction.
N number information is missing in most figures.
Discussion
The main conclusion have been not disscused. Why the hypopneas increase lead to O2 saturation improvement?
The conclusions do not reflect the obtained results. Too speculative.
Reviewer 3 Report
The article "Increasing hypopnea in restricted breathing disturbance improves postoperative oxygen saturation in patients with very severe obstructive sleep apnea" presents clinical observations conducted on patients undergoing surgical procedure (to treat OSA).
The overall interest of the study is limited, as it is an observation of a limited number of patients (27) giving difficult to bring obvious conclusions.
However, the main question of this article is quite challenging is it proposes a paradigm shift in the way we consider 'surgical success'.
Some sentences need English corrections (i.e first sentence of the abstract, hard to understand...)
The 'Introduction' section is very short. It brings the main elements:
L.46 : "More severe O2 desaturations come from apneas, compared to hypopneas"
L.60: "If more severe O2 desaturations come from apneas than hypopneas and the surgery may shift apneas to hypopneas, it is reasonable to hypothesize that increasing hypopnea in AHI improves postoperative O2 saturation"
However, it looks more like an abstract (giving already the conclusions of the study). More explanations about trends, difficulties to treat (and to evaluate) and post operative consequences, should be added to help the reader to follow the interest of the study.
Some details should be added in the "Materials and Methods" section: i.e the time authors waited before new records of AHI, clarification on statistical analyses (did authors check normality of data before using parametric tests?), after how long time after surgery the AHI and mean SpO2 was considered as 'stable'?
"Results" and "Discussion": results are descriptive, with measurements before and after. Authors propose to reconsider what is called "surgical success", according to SpO2 desaturation and number of hypopnea. It could be of great interest to propose/discuss a classification taking in account initial AHI, or evaluation on quality of life by the patients (before-after), impact of BMI, of age...
This study is clearly instructive and interesting for clinicians. Some more discussion, explanations, interpretation would strengthen the overall interest of the paper.
Round 2
Reviewer 2 Report
The manuscript has been significantly improved. Congratulations!
Reviewer 3 Report
The manuscript submitted has received come corrections and additions, and authors have brought answers to all questions they were asked.
The limited number of patients included in this study (with important inter-individual variations) makes it difficult to build as strong scientific proof. On the other hand, the clinical soundness is of prime interest.
Thus, we would recommend to accept the manuscript as it is, aware that they are scientific limitations that are counterbalanced by the pragmatic clinical interest.